# Improving Regional and Teleseismic Detection for Single-Trace Waveforms Using a Deep Temporal Convolutional Neural Network Trained with an Array-Beam Catalog

**DOI:** 10.3390/s19030597

**Published:** 2019-01-31

**Authors:** Joshua Dickey, Brett Borghetti, William Junek

**Affiliations:** 1Air Force Institute of Technology, Wright-Patterson AFB, OH 45433, USA; brett.borghetti@gmail.com; 2Air Force Technical Applications Center, Patrick AFB, FL 32925, USA; william.junek@us.af.mil

**Keywords:** geophysical signal processing, pattern recognition, temporal convolutional neural networks, seismology, deep learning, nuclear treaty monitoring

## Abstract

The detection of seismic events at regional and teleseismic distances is critical to Nuclear Treaty Monitoring. Traditionally, detecting regional and teleseismic events has required the use of an expensive multi-instrument seismic array; however in this work, we present DeepPick, a novel seismic detection algorithm capable of array-like detection performance from a single-trace. We achieve this performance through three novel steps: First, a high-fidelity dataset is constructed by pairing array-beam catalog arrival-times with single-trace waveforms from the reference instrument of the array. Second, an idealized characteristic function is created, with exponential peaks aligned to the cataloged arrival times. Third, a deep temporal convolutional neural network is employed to learn the complex non-linear filters required to transform the single-trace waveforms into corresponding idealized characteristic functions. The training data consists of all arrivals in the International Seismological Centre Database for seven seismic arrays over a five year window from 1 January 2010 to 1 January 2015, yielding a total training set of 608,362 detections. The test set consists of the same seven arrays over a one year window from 1 January 2015 to 1 January 2016. We report our results by training the algorithm on six of the arrays and testing it on the seventh, so as to demonstrate the generalization and transportability of the technique to new stations. Detection performance against this test set is outstanding, yielding significant improvements in recall over existing techniques. Fixing a type-I error rate of 0.001, the algorithm achieves an overall recall (true positive rate) of 56% against the 141,095 array-beam arrivals in the test set, yielding 78,802 correct detections. This is more than twice the 37,572 detections made by an STA/LTA detector over the same period, and represents a 35% improvement over the 58,515 detections made by a state-of-the-art kurtosis-based detector. Furthermore, DeepPick provides at least a 4 dB improvement in detector sensitivity across the board, and is more computationally efficient, with run-times an order of magnitude faster than either of the other techniques tested. These results demonstrate the potential of our algorithm to significantly enhance the effectiveness of the global treaty monitoring network.

## 1. Introduction

Adherence to the comprehensive nuclear test ban treaty is currently verified by the detection, location and identification of seismic events, often at regional (>200 km) and teleseismic distances (>2000 km). Automated seismic detection is the critical first step in this process, and it is imperative that the events be detected by multiple stations, as this increases the overall accuracy of the final location estimate. As such, maintaining a large network of highly sensitive seismic detectors is key to the treaty monitoring community [1,2].

Traditionally, sensitive teleseismic detection has required the use of a multi-instrument seismic array, a strategy which dates back to the Geneva Conference of Experts in 1958 [3]. The sensitivity is achieved through beamforming [4], a spacial filtering technique that relies on a tuned network of interconnected seismometers which form a single station. This technique is extremely effective; however, it is quite expensive to implement due to the additional sensors and processing required, and unfortunately, beamforming is inapplicable to single-instrument stations. As such, the vast majority of seismic stations around the globe are simply unable to detect weak regional and teleseismic events.

In this work, we create an automatic detector with array-like performance from a single trace, capable of detecting these signals which were previously too weak to detect with a single sensor. Building on several recent efforts which apply the power of convolutional neural networks to the detection of local events [5,6,7], we apply similar techniques to the detection of regional and teleseismic events, events previously only detectable using a seismic array. Specifically, we tackle the following research objective: Using the analyst reviewed catalog of events from an array-beam as the reference, and fixing a type-I error rate of 0.001, create a transportable single-trace detection algorithm with improved recall over existing detectors.

To tackle this objective, we present DeepPick, a single-trace automatic detection algorithm capable of detecting up to 80% of the events in an array-beam catalog. The algorithm is based on a deep Temporal Convolutional Neural Network (TCN), and it is trained against more than five billion raw seismic samples containing 608,362 labeled seismic arrivals from seven array-beam catalogs in the International Monitoring System (IMS) network: TXAR, PDAR, ILAR, BURAR, ABKAR, MKAR and ASAR located in Lajitas Texas, Pinedale Wyoming, Eielson Alaska, Bucovina Romania, Akbulak Kazakhstan, Makanchi Kazakhstan and Alice Springs Australia, respectively. Performance is reported by training the algorithm against five years of data from six of the arrays, and testing it against a full year of data from the seventh, remaining array. All seven arrays are tested in this manner, resulting in recall rates ranging between 50% and 80% against the individual array beam catalogs, and an overall average of 56% against the combined catalog. This represents a marked improvement over the performance of existing algorithms for the same task. For example, we deploy both a modern adaptation of the STA/LTA detector (FBPicker) [8] and a kurtosis-based detector (KTPicker) [9] against the same test set, achieving only 27% and 42% detection rates respectively. Additionally, our algorithm is highly computationally efficient, demonstrating an order of magnitude reduction in computation time over both of the other algorithms.

While there have been several recent efforts to employ convolutional neural networks for seismic detection, our effort here differs in three significant ways. First, our detector was trained and tested using a higher-fidelity reference catalog with an 8 dB improvement in sensitivity over traditional catalogs, which we accomplished by using an array-beam catalog as a reference. Second, whereas previous efforts treat detection as a binary classification problem (thus requiring a secondary algorithm for arrival time picking), our algorithm follows the traditional seismic detection approach of first creating a characteristic function. As such, we show that our effort follows very much in-line with traditional methodologies, but with significant quantitative improvements. Third, our detector is the first to focus on teleseismic detection, a task which depends on recognizing long-period features, and which we accomplish using a Temporal Convolutional Neural Network (TCN) with a wide receptive field. As such, we present three major contributions to the literature:A unique training technique for single-trace detection algorithms, which uses array-beam catalogs as a high-fidelity referenceA novel training objective, exponential sequence tagging, which trains the TCN to transform single-trace waveforms into an ideal characteristic function with weighted exponential peaks at predicted arrival timesDeepPick: a single-trace detection algorithm capable of achieving array-level detection performance

In the remainder of this work, we provide context for and explain these contributions by first reviewing the related literature, then outlining our methodology, and finally detailing and discussing our results.

## 2. Related Work

Automatic seismic detection algorithms are a key component of any modern seismic network, and here we review the literature pertaining to this important field. Our review begins with a discussion of the traditional detection algorithms, then investigates teleseismic detection in particular. Finally, this section provides a detailed examination of the nascent field of convolutional neural network-based detectors, while emphasizing the gaps in the research we intend to address in our own work.

### 2.1. Traditional Seismic Detection

Traditional algorithms for seismic signal detection usually share a simple, common framework: A comparison is made between the current value of the seismic signal (or some function of it) and a predicted value, and a detection is declared whenever this comparison exceeds some factor. From this simple concept has arisen a vast number of algorithms, which vary primarily based on their choice of the function to which the detection is applied. This function is often referred to as the characteristic function (CF) of the algorithm [10].

By far the most common traditional technique for seismic signal detection is the short-term average, long-term average (STA/LTA) detector, first described by Freiberger [11]. In its simplest form, this technique employs a bandpass filter to compute the characteristic function, with the predicted value equal to the long-term average and the current value equal to the short-term average. The current and predicted values are then compared via a ratio which is then subjected to some static threshold, as detailed in Figure 1. Numerous adaptations and enhancements to this STA/LTA detector have been proposed, most notably by Allen [12] and Baer [10], who increased detection efficiency by employing novel characteristic functions based on a combination of the signal and its time derivatives. More recently, modern iterations of the STA/LTA algorithm have employed multiple characteristic functions across multiple frequency bands with great success [8].

Unfortunately, the STA/LTA family of algorithms have an inherent difficulty identifying events that emerge from a noisy pass-band [13]. Fortunately, unlike random noise, seismic signals have higher-order statistics (such as skewness and kurtosis) which are non-zero [14]. This means that the signal and noise energies can be well-separated using characteristic functions based on these higher-order statistics (HOS), which serve as the basis for another common subset of seismic signal detectors, the HOS-based detectors described in [13,15,16]. These algorithms can provide excellent performance, but tend to be more computationally expensive.

Other more exotic characteristic functions that have enjoyed success include variations of the Walsh transform [17] and the wavelet transform [18]. Furthermore, there are families of algorithms used to determine the precise arrival time after a detection has been made. These are commonly referred to as autoregressive methods, which employ various techniques, the most common of which was proposed by Sleeman [19] and uses the Akaike Information Criterion.

### 2.2. Teleseismic Detection

Having examined seismic signal detection in general, we now turn our attention specifically to the literature concerning the detection of regional and teleseismic signals. Such signals can be particularly challenging to detect, as their signal strength is often significantly attenuated by the longer path of travel. To address this, one of the most successful techniques for regional and teleseismic signal detection is Beamforming [1,20], introduced in [21]. Beamforming gains its effectiveness by linearly combining signals from multiple sensors according to the estimated arrival direction, also known as the back-azimuth, allowing it to pick out signals beneath the noise floor of a single sensor [4]. Unfortunately, beamforming requires an interconnected array of seismometers, spread out across a large geographical area measuring tens or even hundreds of kilometers. An example array layout, along with a demonstration of the beamforming technique is detailed in Figure 2.

Another outstanding technique for the detection of weak teleseismic events is the phase-matched filter [22] popularized by [23,24] in the early 1990s. These pattern matching techniques are a type of Empirical Signal Detector that work by comparing incoming seismic waveforms to canonical examples in the extant seismic record [25,26]. They are particularly effective for the detection of highly correlated repeating events, even for very weak magnitudes [27]. Unfortunately, to date, this technique is not generally applicable, as only 18% of all global events possess sufficient similarity to be detected with this technique [28].

### 2.3. Seismic Detection with Convolutional Neural Networks

Convolutional Neural Networks (CNNs) are revolutionizing the science of signal processing from computer vision to speech recognition, and they are poised to do the same for seismic signal processing as well. This begs the question: why are CNNs so effective at signal processing tasks? To answer this, we note that at their core, CNNs are comprised of a set of digital filters which are convolved with the signal, where the optimal filter weights are learned by applying stochastic gradient descent across some objective function. In effect, the CNN can quickly explore a wide range of filters and empirically converge on ones that work well. The real power of CNNs comes from the ability of the network to learn many filters simultaneously, combine their outputs with non-linear activations, and then feed these activations into further layers of learned filters, ultimately allowing the CNN to learn complex non-linear transformations. This structure allows CNNs to accomplish a wide range of signal processing tasks in a way that is actually quite similar to the traditional analyst-driven methods, except for the fact that the empirical search for the optimal filter weights is performed by a computer in much less time and at a much larger scale. The key to learning the optimal transformation has sufficient quality and quantity of labeled training data for the objective function. In addition, due to the vast quantity of labeled seismic data available, seismology is poised to take advantage of the power of CNNs.

Several recent efforts have already been made to apply Deep Convolutional Neural Networks to seismic signal detection. Although this research is still in its infancy, early results have shown great promise.

In [5], the researchers use a convolutional neural network architecture to perform detection on local seismic signals, formulating the task as a binary classification problem. Their dataset was obtained from two seismic stations in the Oklahoma Geological Survey, consisting of 10-s windows with binary class labels: Positive windows were centered around seismic arrival times obtained from an analyst-reviewed arrival catalog, and negative windows were carefully selected to contain no arrival. Against their hold-out test set, they report 100% recall with a high type-I error rate of 1.4%. These results are outstanding, but the most interesting finding in their research comes from their examination of the false positives detected by their algorithm. By applying a correlation detector to their reported false positives, they determined that a substantial portion of these were actually real detections of very weak events. This means that the algorithm learned to detect events below the detection threshold of the catalog on which it was trained. This work highlights the danger of using conventional catalogs to train such a sensitive detector. Additionally, two major limitations exist in this work. First, because of the extreme care taken to produce ‘clean’ noise windows in the test set, their reported type-I error rate is not realistic for operational use. Second, their algorithm is applicable only to local events; the short time windows used (10 s) limit the algorithm’s potential to detect the longer-period (100 s) features characteristic of teleseismic signals.

In [29], the researchers also use a deep CNN to perform seismic signal detection on local events. Their dataset consisted of 4.5 million 4-s windows of waveform data recorded and classified by the Southern California Seismic Network. Their task was formulated as a classification problem, assigning one of three classes to each window, P-wave (primary phase arrival), S-wave (secondary phase arrival) and noise. This resulted in 1.5 million windows containing a P-wave arrival, 1.5 million windows containing an S-wave arrival and 1.5 million windows including no arrival. Their validation set consisted of a randomly sampled 25% of the overall data, resulting in 1.1 million seismograms evenly split between the three classes. On the validation set, they report a recall of 96% at a type-I error rate of less than 1%. These results are very impressive, and show that the convolutional neural network is capable of achieving state-of-the-art performance on the seismic signal detection task. A limitation of this work is that it is applicable only to local signals; the researchers only report recall for signals originating within 100 km of the recording station.

In [6], the same research team considers arrival time estimation. Here they formulate the task as a regression problem, and consider only 4-s windows of data, centered around an arrival, with up to half a second of variance in the arrival time from the center of the window. For this task, they report a mean average error of less than 0.02 s from the analyst-recorded picks. Once again, these signals are limited to local events.

Seismic signal detection is an active area of research, with new, improved algorithms being developed capable of achieving near-perfect accuracy for local event detection. Despite this, little effort has been made to extend detection to regional and teleseismic events without the use of a seismic array. This is the research objective our work shall address.

## 3. Materials and Methods

The research objective is to build a single-trace detection algorithm capable of detecting weak regional and teleseismic signals with array-like performance. We know that such detections are possible using a full seismic array and we have seen the potential for achieving such detections using a deep neural network. Our approach is to employ a deep TCN model, feed it a single-trace input sequence, and train it to produce a characteristic function with distinct peaks centered on arrival times obtained from an array beam catalog. In this section, we explore this approach in detail, first defining our dataset, and then describing our modeling strategy.

### 3.1. Data Collection

The success of any deep neural network algorithm lies largely in the careful collection and construction of the training data. This subsection presents a dataset suitable for training a deep seismic detection algorithm. In particular, it details two of our major contributions: First, a description of a novel method for obtaining a high-fidelity dataset of single-trace waveforms with labeled arrival times below the noise floor. Second, it presents exponential sequence tagging, the unique sequence-to-sequence modeling schema used to create an ideal characteristic function for picking arrival times. This subsection concludes with the details of training, test and validation datasets.

#### 3.1.1. High Fidelity Arrival Catalog

At first glance, obtaining a dataset for training a seismic detector would appear to be trivial, as analyst-reviewed arrival catalogs are freely available for millions of seismic events at thousands of seismic sensor elements. Unfortunately, despite the rigorous review process and the extensive cross-referencing, each single-trace arrival catalog only contains picks for signals with sufficient strength to be conventionally detectable from within that trace. This is a significant limitation when the goal is to train a detector more sensitive than the conventional one. Fortunately, there are certain sensor elements with accurate cataloged arrival times for regional and teleseismic signals below the noise floor; namely, any sensor element located at the reference point of a seismic array (usually a broadband 3-channel instrument). Using conventional methods, this ’reference-element’ alone is unable to make accurate detections for sub-noise floor events; however, the array beam as a whole can make these detections very accurately [20], and the beam arrivals are conveniently aligned to indicate arrivals at this reference element. Thus, by obtaining singe-trace input data from the reference element, and by obtaining labeled arrivals times from the array beam, we can create a labeled single-trace dataset with signals below the noise floor. As an example, Figure 3 demonstrates the significant 8 dB improvement in detector threshold provided by the Makanchi Array beam in eastern Kazakstan.

For future researchers interested in establishing a similar high-fidelity dataset, we provide here a four step process:**Step 1: Obtain the Array-Beam Catalog** Arrival-time catalogs can be downloaded through a web query of the International Seismological Centre (ISC) Bulletin for seismic arrivals (http://www.isc.ac.uk/iscbulletin/search/arrivals/), by specifying the desired array station name (i.e., MKAR)**Step 2: Identify the Array-Beam Reference Point** The array-beam reference point coordinates can be found through a web query of the ISC station registry (http://www.isc.ac.uk/registries/search/), by again specifying the desired array station name.**Step 3: Identify the Array-Beam Reference Elements** Available reference elements can then be found by a second web query of the ISC station registry, using the reference point coordinates as the search criteria. For the MKAR array, there are two sensor elements located at the reference point: MK31 and MK32.**Step 4: Obtain Reference Element Waveforms** Raw waveforms can be downloaded from the Incorporated Research Institutions for Seismology (IRIS) Database using ObsPy.

#### 3.1.2. Idealized Characteristic Function: Exponential Sequence Tagging

With established high-fidelity sources for both waveforms and arrival times, the next step is to generate input/output pairs for training a seismic detector. Most previous efforts to build an ML-based seismic detector have been formulated as a binary classification task; the input data is partitioned into fixed length windows, each paired with a single Boolean class label: positive class labels are assigned to windows where a signal is present and negative class labels are assigned to windows where signal is absent. This traditional formulation is convenient, as the classes can easily be balanced at training time and it is the common method employed in most recent works in the literature [5,29,30]. However, this methodology has three major limitations: First, it is not ideally suited for real-time processing, as the algorithm needs access to a signal window several seconds beyond the signal arrival. Next, it requires a secondary algorithm applied within the detection window, to estimate the precise arrival time [6]. Finally, this methodology is not well suited for the detection of regional and teleseismic signals. Teleseismic signals are characterized by long-period features with frequency components as low as 0.01 Hz [31], and the detection of these features necessitates windows that are several minutes in length. Unfortunately, this resolution is far too coarse for classification, and often covers multiple arrivals in a single window. As such, there are two conflicting requirements for creating binary classification windows in a teleseismic detection dataset:Input windows must contain many samples to capture long-period teleseismic featuresOutput labels must cover a few samples to allow meaningful temporal resolution for the detection windows

To resolve this conflict, we reformulate the task. Instead of performing binary classification on each window, we perform regression on each sample, which is known as sequence-to-sequence modeling [32]. Each training window is labeled with an output sequence of real-valued numbers; each sample in the input sequence is assigned a corresponding value in the output sequence. Coincidentally, this process is nearly identical to the generation of the characteristic function in traditional seismic detector algorithms. The difference is that whereas traditional algorithms specify the transformation in order to produce a characteristic function that has defined arrivals, our algorithm can specify the characteristic function explicitly and let the neural network learn the transformation. As such, we can assign labels corresponding to any idealized characteristic function we desire. However, what labels should we assign? A naive formulation is to simply assign a ‘one’ at each cataloged arrival time and assign a ‘zero’ everywhere else. This characteristic function would essentially look like a delta function at each cataloged arrival. This formulation is called sequence tagging [33], and it works well for relatively balanced classes [34]. Unfortunately, binary sequence tagging does not work well for teleseismic detection, as it results in an extreme class imbalance of several orders of magnitude, which hinders learning.

For this work, we present a novel formulation which we call exponential sequence tagging. This formulation produces a characteristic function that consists of a mirrored-exponential function applied at each cataloged arrival time, as shown in Figure 4b. To be precise, the labels in the output sequence are nominally zero until a cataloged arrival time, at which point they increase and decrease exponentially, according to the mirrored exponential decay function given in Equation (Equation 1), where λ is the decay rate. This characteristic function is quite similar to that used in the ‘suspension bridge’ seismic detection algorithm, proposed in [35] and referenced in [36].
(1)$$\begin{array}{c} y\left(t\right)=e^{-\lambda |t|} \end{array}$$

Because each leg of the mirrored exponential decay function is both monotonic and deterministic, the value at each non-zero label can be used to directly infer the precise arrival time. In addition, because the algorithm learns to match these labels with its output, every non-zero sample in the output is effectively an arrival time estimation. With this in mind, we assign one additional computation to our algorithm at run-time: a cross-correlation of the predicted output sequence with the original exponential decay function. This filters the output and effectively aggregates the arrival time estimates for an even more precise arrival time pick. Because the height of the resulting peak is the correlation between the network model’s output and the original exponential, it represents the certainty that the peak is a true arrival and can be used to set the threshold of the detector. Figure 4c,d shows an example of the predicted output, both before and after this cross-correlation is applied, where (d) depicts the final characteristic function.

#### 3.1.3. Training, Validation and Test Sets

Using this approach to build our training dataset, we obtained a catalog of all local, regional and near-teleseismic arrivals for the seven array beams during a five year period from 1 January 2010 to 1 January 2015. We generated this catalog through a web query of the ISC Bulletin for seismic arrivals which can be accessed here: http://www.isc.ac.uk/iscbulletin/search/arrivals/. The corresponding waveforms were then windowed around each arrival (the windows were 6 min in total length, sampled at 40 Hz for a total of 14,400 samples per window), and the raw traces were pulled from the IRIS Database, for the vertical channel of the nominal seismometer for each array (PD31_BHZ, TX31_BHZ, IL31_BHZ, MK31_BHZ, ABK31_BHZ, BUR31_BHZ and AS31_BHZ). This was accomplished via a custom Python script based on ObsPy-1.1.0 [37], and yielded a dataset of 608,362 picks out of a total training size of more than five billion samples. The only pre-processing applied to the raw data was a normalization, detrending and bandpass filtering between 0.02 Hz and 10 Hz.

From this training dataset, we selected one month of data from each array (1 January 2010 to 1 February 2010), as a validation set. This validation set was used to tune the models, with final model selection based on validation set performance.

To build our testing dataset, we also obtained a catalog of all local, regional and near-teleseismic arrivals for the seven array beams, in this case during a one year period from 1 January 2015 to 1 January 2016. This test includes 141,095 arrivals in the seven array beam catalogs. This test set data was not used to train or tune the models, only to report performance against each array. Additionally, to ensure that our reported performance figures are indicative of the expected performance against novel stations, we actually trained seven separate models, each on a different partition of six arrays and tested against the seventh, such that performance for all seven arrays is reported using a model that did not have access to any training data from that array, demonstrating the generalizability and transportability of our detector.

### 3.2. Modeling

Now that we have defined our dataset, we present a description of our modeling methodology, detailing the model architecture, hyper-parameter search vectors, and evaluation metrics.

#### 3.2.1. Model Architecture

Our model architecture is based on the Temporal Convolutional Network. TCNs are deep convolutional architectures characterized by layered stacks of dilated causal convolutional filters with residual connections [38]. These characteristics offer several distinct advantages for a seismic detection algorithm, which we briefly summarize:Residual connections allow the model to have high-capacity and stable trainingCausal convolutions allow the model to make predictions on continuous streaming trace dataDilated convolutions allow precise control over the receptive field

The receptive field is of primary importance for time-series modeling, as it explicitly limits the learnable feature periodicity at a given layer. As such, one of our key design parameters was to ensure adequate receptive field for our algorithm. The equation for calculating the receptive field for a given convolutional layer, *l*, and dilation rate, *d* is given in (Equation 2):(2)$$\begin{array}{c} rField(l)=rField(l-1)+[kernelSize-1]*d \end{array}$$

Using this equation, the network is designed to have a receptive field of roughly 100 s (or 4000 samples), allowing it to learn long-period features down to 0.01 Hz. This is accomplished in just 4 layers, as shown in Table 1. Another key design parameter was to ensure that the dilation rate in each layer remained less than the receptive field in the previous layer to prevent gaps in receptive field coverage. Notice that this constraint is maintained even for the final layer with a dilation rate of 256, as the previous layer had a receptive field of 331. The model architecture is shown in Figure 5.

#### 3.2.2. Hyper-Parameter Search Vectors

Fixing this basic architecture, we engage in a limited hyper-parameter search over two general vectors: the optimal shape for the exponential function, and the optimal capacity for the neural network.

Optimization over the decay rate of the exponential was varied across 3 choices, {0.015, 0.02, 0.04}, selected based on visual inspection. Optimization over model capacity was conducted across two parameters, number of stacks and number of filters. Each parameter was varied across 4 choices, {2, 5, 9, 12} and {5, 10, 15, 20} respectively, ranging from a minimal capacity network (2 stacks with 5 filters and only 3517 parameters) to a high capacity network (12 stacks with 20 filters and 328,681 parameters). Because these two parameters are highly interrelated, the search was conducted exhaustively, for a total of 16 models. The final hyper-parameter selections were based on validation loss curves.

### 3.3. Evaluation Criteria

The research objective is to determine the maximum achievable recall of our single-trace detection algorithm against the array beam catalogs. Because recall is a classification metric, and because the task is a regression problem, the next step is to define the method for calculating recall.

Each detection window is 4 s, identical to the window length used in [6]. The number of Total Positives is the number of labeled arrivals in the dataset, and the number of Total Negatives is the number of windows (length of the dataset in seconds divided by 4) minus the number of Total Positives, which is a conservative estimate. A predicted arrival is any peak in the output sequence with a value above a threshold. A True Positive is any predicted arrival within 2 s before or 2 s after a labeled arrival, and a False Positive is any predicted arrival not within 2 s before or after a labeled arrival. A False Negative is a labeled arrival not within 2 s of any predicted arrival, and thus the count of True Negatives is the Total Negatives minus False Negatives. From these definitions, standard equations are used (Equation 3) to calculate recall (true positive rate) and Type-I error (false positive rate):(3)$$\begin{array}{cc} \mathrm{Total}\;\mathrm{Windows} & =\frac{\mathrm{Dataset}\;\mathrm{Length}}{\mathrm{Window}\;\mathrm{Length}}\\ \mathrm{Total}\;\mathrm{Positives} & =\#\mathrm{of}\;\mathrm{Cataloged}\;\mathrm{Arrivals}\\ \mathrm{Total}\;\mathrm{Negatives} & =\mathrm{Total}\;\mathrm{Windows}\;-\;\mathrm{Total}\;\mathrm{Positives}\\ \mathrm{Recall} & =\frac{\mathrm{True}\;\mathrm{Positives}}{\mathrm{True}\;\mathrm{Positives}\;+\;\mathrm{False}\;\mathrm{Negatives}}\\ \mathrm{alpha} & =\frac{\mathrm{False}\;\mathrm{Positives}}{\mathrm{Total}\;\mathrm{Negatives}} \end{array}$$

Using these definitions, and treating the analyst-reviewed array beam catalogs as ground truth, performance is reported in terms of both receiver operating characteristic (ROC) curves and recall. When reporting recall, a type-I error rate of 0.1% is used. It should be noted that this is an order of magnitude lower than the error rate reported in [5,29,30], as it is more appropriate for operational use. Because the primary goal is weak-signal detections, recall is also reported as a function of signal to noise ratio (SNR). SNR is defined as the log ratio between the short-term and long-term average power, as given in Equation (Equation 4), with a short-term window consisting of 5 s after the arrival, a long-term window consisting of 40 s before the arrival, and a bandpass filter applied from 1.8 to 4.2 Hz.
(4)$$\begin{array}{c} \mathrm{SNR}=10*\log_{10}\left(\frac{PWR_{STA}}{PWR_{LTA}}\right) \end{array}$$

Additionally, in order to asses the value of our algorithm over existing single-trace methods, performance is compared against two common automatic detectors, FBPicker [8] and KTPicker [9]. These detectors are implemented in the PhasePApy [39] package for python, which was developed by the Oklahoma Geological Survey, and has been in operational use there since 2015. The three algorithms are then compared by detector efficiency, arrival time estimation, and overall computation time. Details of the implementation of these two algorithms can be found in Appendix A.

## 4. Results

Two hyper-parameter search vectors were optimized in the model: exponential decay and model capacity. As shown in Table 2, decay rates between 0.015 and 0.040 were explored, and a decay rate of 0.020 yields the highest recall on the validation set.

Fixing the decay rate at 0.020, the overall capacity of the model is varied by increasing both the number of residual stacks, *s*, and the number of 1D convolutional filters, *f*. Total training time for each model was approximately 200 h on an Nvidia GTX 1080 Ti, and the results of this search indicated that model capacity is optimized with 12 stacks and 15 filters, as increasing capacity beyond this point appears to have marginal value. This yields a final model with 12 residual stacks as shown in Figure 5, with 15 filters on each 1D convolution, for a total of 185,311 fully convolutional parameters.

Table 3 shows the results of evaluating the final model against the hold out test set. Across the seven arrays, the detector is able to correctly classify 56% of the 141,095 array beam picks, yielding 78,802 correct detections. This is a 35% improvement over the 58,515 detections found by the KTPicker, and more than double the 37,572 detections found by the FBPicker for the same period.

The ROC curves shown in Figure 6 further illustrate the performance of the algorithm. It should be noted that the type-I error rate of approximately 0.001 represents performance to the left of the elbows of the ROC curves and sub-optimal detector efficiency. For pure academic exercise, a much better choice would be to relaxing the type-I error rate to 0.01, as observed in other recent works [5,29]. This increases the overall recall of DeepPick to 77%. Unfortunately, such a large type-I error rate is not acceptable for operational use, as it represents far too many false positives for a human analyst to deal with. Appendix B presents the performance of the algorithm on several example waveforms, comparing it to FBPicker and KTPicker.

The objective is to detect weak, distant events. This requires a detector with enough sensitivity to pick out signals near the noise floor. To explore the algorithm’s performance at this task, its ability to detect signals with very low signal to noise ratio is evaluated. Using the array beam catalog as a baseline, Figure 7 depicts recall as a function of SNR. This demonstrates that DeepPick maintains a more than 90% recall for signals with an SNR of at least 10 dB for each of the seven arrays in the test set. Signals with an SNR of 10 dB or below are quite difficult to detect from a single trace, as evidenced by the dashed lines in the plot, which represent the detections two other detection algorithms, FBPicker and KTPicker. These graphs indicate that DeepPick maintains at least a 4 dB advantage in sensitivity over both of the other detection algorithms across all seven arrays.

Finally, we report the algorithm’s performance for the arrival time estimation task as detailed in Table 4 (Arrival time error, Δt, is only reported for true positives (Δt<2 s)). Here, the algorithm achieves a mean average error of 0.45 s from the analyst picked arrival times, with a distribution detailed in Figure 8. This plot shows that while the most common histogram bin corresponds to an absolute error of less than 0.025 s, the weakest signals are frequently missed by more than a second. This error is on par with other automatic detectors as shown in Table 4.

## 5. Discussion

The results in Table 3 demonstrate that the DeepPick algorithm is capable of achieving a recall of between 50 and 80% against the analyst-reviewed picks from seven array-beam catalogs with a type-I error rate of 0.001. The low end of this range, 49% recall at ASAR, represents a significant improvement over the performance of existing single trace algorithms (25% and 37% for FB and KT respectively). However, the spread in results is quite large, and suggests the need to examine the underlying cause of this performance variance.

The two stations with the worst performance are ILAR and ASAR. Interestingly, these two stations also use a different sensor, the Guralp CMG-3TB, from the other five stations, which all use the Geotech KS54000. This shows the importance of training the algorithm on stations with the same instrument type as the stations for which the algorithm is intended to be deployed against operationally. The two stations with the best results are ABKAR and BURAR. Interestingly, due to higher noise levels at these sites, the array catalogs for these two stations contain relatively fewer events with relatively larger magnitudes. This makes the detection of these events easier, and the recall rates of 74% and 80% reflect this fact. PDAR, TXAR, and MKAR use a common instrumentation, share similar geology and have similar noise levels; as expected, they also share similar recall rates of 54%, 57% and 61% respectively.

The computational efficiency of our algorithm is measured in run-time (seconds) required to build an automatic catalog across a full year of data. Table 5 shows that DeepPick has an order of magnitude increase in computational efficiency over the FBPicker and more than two orders of magnitude increase over the KTPicker. It should be noted that the implementations of FBPicker and KTPicker used here are actual operational implementations used by the Oklahoma Geological Survey. This illustrates the extreme efficiency of the DeepPick algorithm.

These results show that the primary determinant of algorithm success lies in the degree of similarity between the training stations and the testing station. As such, when deploying this algorithm for operational use it is important to find suitable arrays to train on in order to maximize performance. In any case, the algorithm shows decent performance even when trained across different geographical areas and sensor types.

## 6. Conclusions

Weak teleseismic event detection is normally only possible using an array of seismic instruments and sophisticated processing techniques. Even recent works in the literature make little attempt to extend single-trace detection algorithms beyond local events. This is primarily due to the lack of available training data, an issue which we address by mining the seismic catalogs in a unique way, building our catalog for an array beam while taking our event waveforms from a single array element. Using this training data, temporal convolutions and a unique exponential sequence tagging function we develop a powerful tool for weak signal teleseismic detection. The DeepPick algorithm is able to accurately detect twice the number of events detected by the STA/LTA algorithm commonly used, and does it significantly faster.

The findings in this work represent an important step forward in the field of teleseismic detection, demonstrating that accurate teleseismic event detection is possible from a single seismic instrument. The DeepPick algorithm has the potential to open up thousands of additional automatic detections to single-instrument seismic stations each year, without the need for additional sensors and equipment.

There is still potential for much improvement. In this work, we develop a single-trace detector, applied only to a single channel of data from a three channel instrument; future work could extend our results to include data from all three channels of the instrument. Furthermore, an application of the same technique to an entire array of channels could also prove interesting, and the potential exists to improve our results significantly by incorporating more channels of data. Additionally, the focus of this work has been primarily centered on producing a detector with increased sensitivity and recall, whereas future work could focus on using similar techniques to produce a detector with an even lower false positive rate.

## Figures and Tables

**Figure 1 sensors-19-00597-f001:**
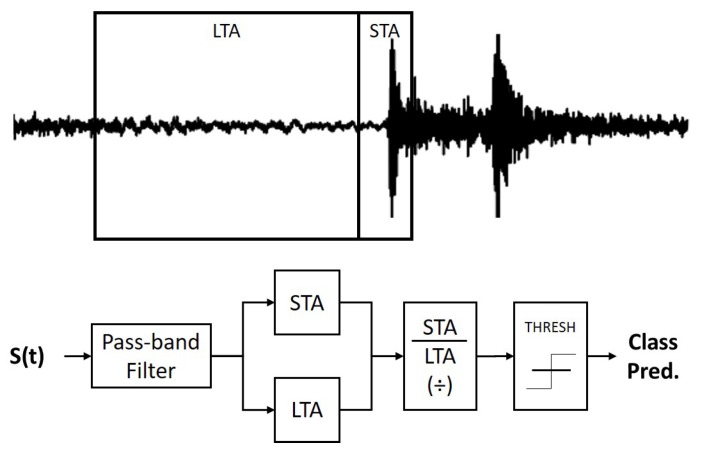
**Top:** Example seismic waveform, annotated to show the STA and LTA windows. **Bottom:** Diagram detailing the operation of the STA\LTA algorithm.

**Figure 2 sensors-19-00597-f002:**
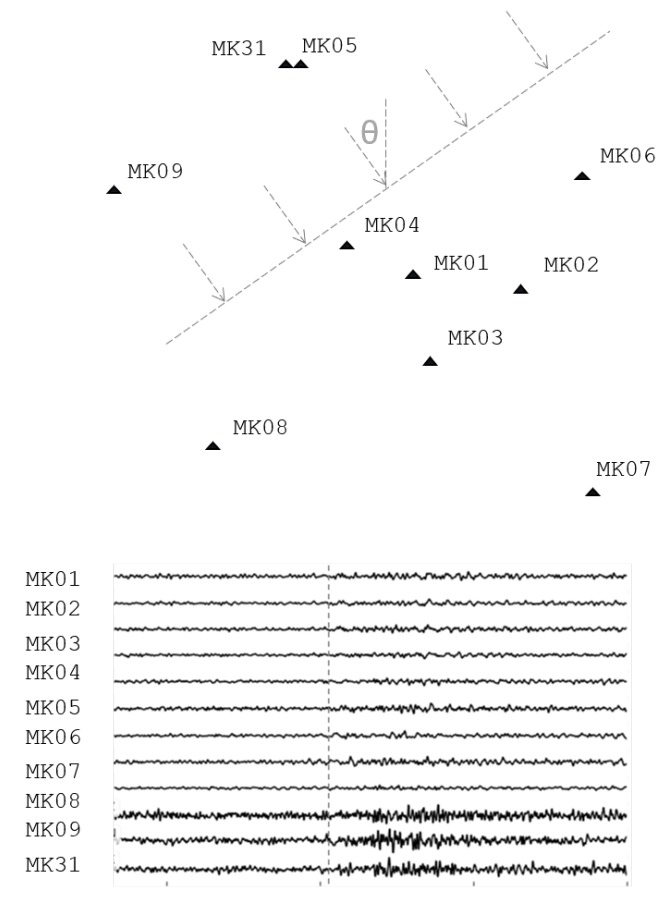
**Top:** Layout of the 10-element Makanchi Seismic Array, MKAR, in eastern Kazakhstan. The dashed lines illustrate an incoming teleseismic wave with calculated back-azimuth, θ. **Bottom:** Seismic waveforms from an arriving teleseismic event. Beamforming aligns these waveforms via the back-azimuth and wavefront velocity, and then linearly combines them to yield a higher SNR, improving the detection threshold significantly.

**Figure 3 sensors-19-00597-f003:**
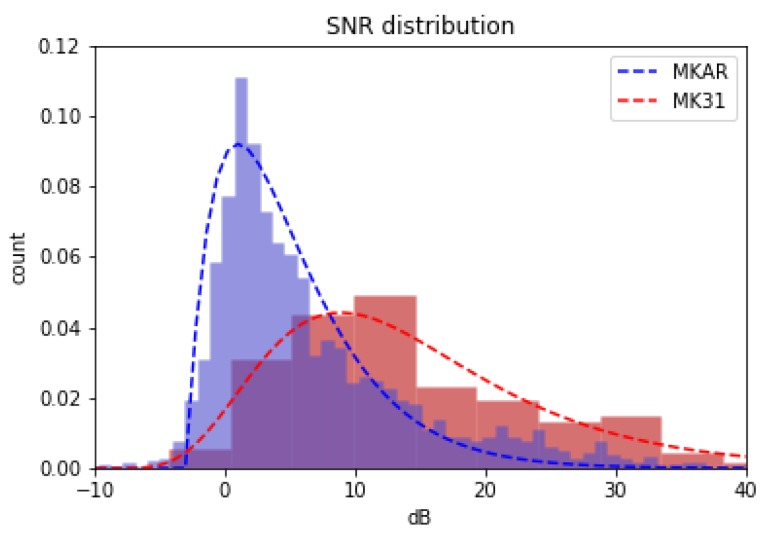
Normalized histograms showing the SNR distributions of detected signals from two seismic arrival catalogs. Both catalogs contain detections for the exact same location, MK31, which is the reference element of the MKAR seismic array. The MK31 catalog is based on a single-trace detection algorithm applied to the MK31 instrument alone, while the MKAR catalog is based on beam-formed picks from the entire 10-instrument array. The mean SNR detected by the array beam is 8 dB lower than that of the single-trace. This lower detection threshold results in nearly an order of magnitude more detections in the MKAR catalog compared to the MK31 catalog.

**Figure 4 sensors-19-00597-f004:**
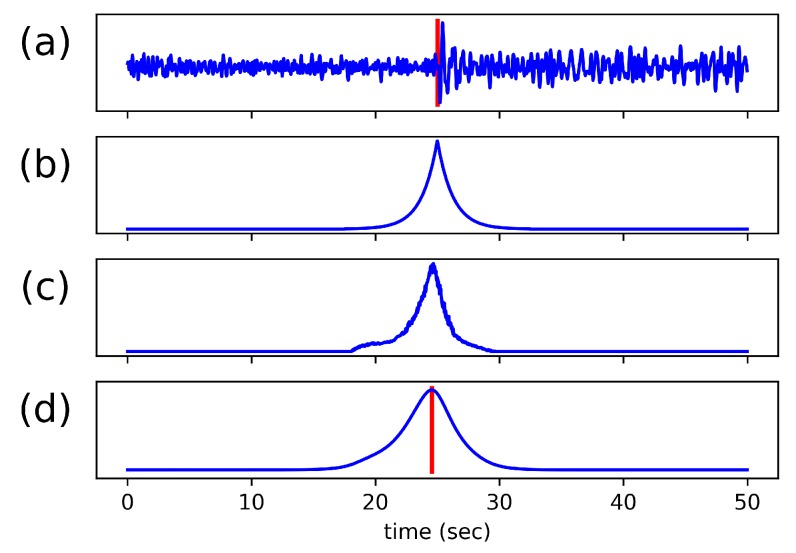
(**a**) Input Sequence containing an arrival marked in red (**b**) Labeled output sequence using the exponential function. (**c**) Predicted output sequence from the model. (**d**) Cross-correlation between the predicted output sequence and the exponential function. The predicted arrival is marked in red.

**Figure 5 sensors-19-00597-f005:**
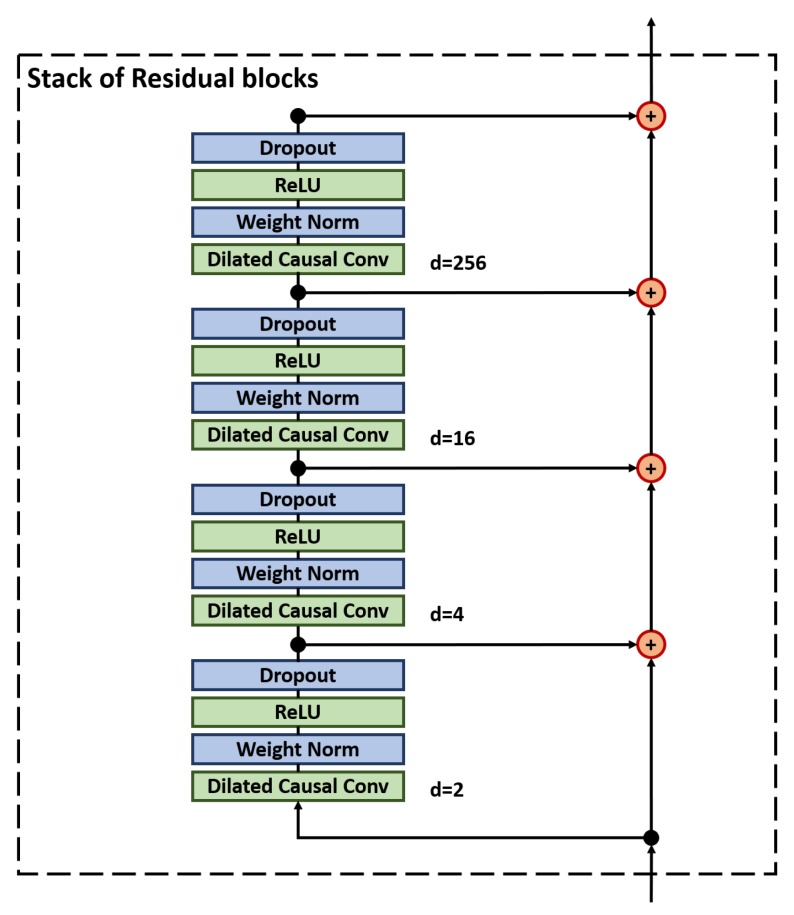
One stack of our chosen TCN architecture. As shown, the stack consists of four separate layers of convolutional filters, which are progressively dilated to provide a wide receptive field. The number of filters in each layer and the overall number of stacks are two hyper-parameters that determine the overall model capacity. As shown, each layer uses a Rectified Linear Unit (ReLU) activation function, and employs two forms of regularization: weighted normalization and Dropout.

**Figure 6 sensors-19-00597-f006:**
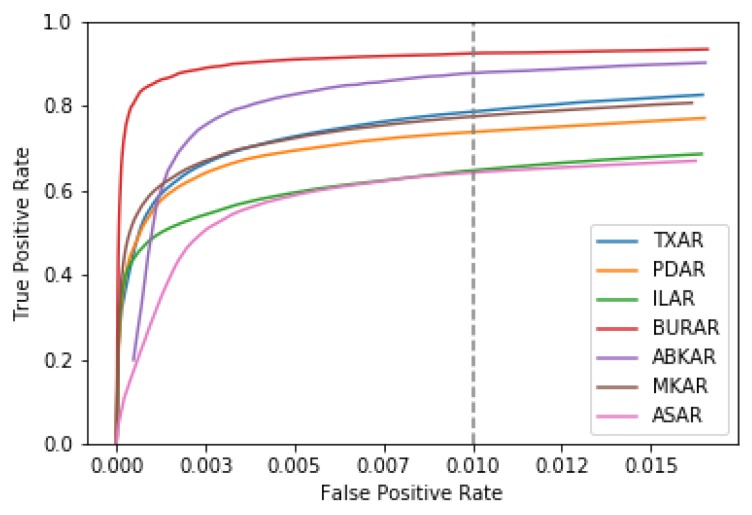
Receiver Operating Characteristic Curves for each of the seven arrays in the hold-out test set. A dashed line is shown in grey, indicating an alpha of 1%.

**Figure 7 sensors-19-00597-f007:**
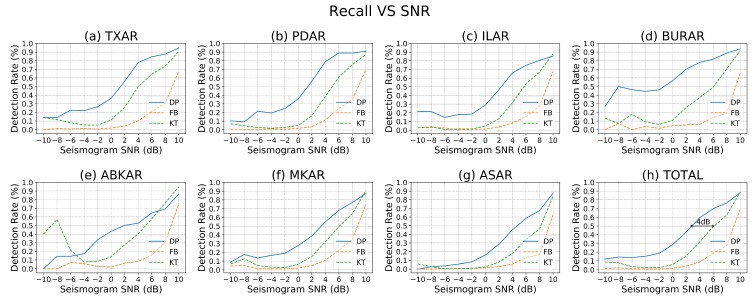
Test-set Recall, reported as a function of SNR, at a fixed type-I error rate of approximately 0.001. Results are compared directly between the three algorithms, DeepPick (DP), FBPicker (FB), and KTPicker (KT). Please note that several of the reference catalogs contain fewer arrivals below −8 dB SNR, resulting in some irregularities to the far left of the plots.

**Figure 8 sensors-19-00597-f008:**
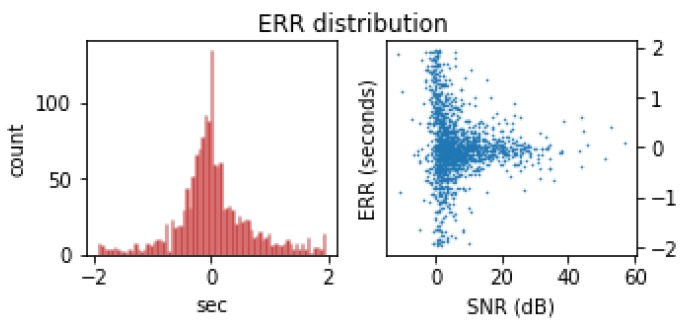
Residual analysis on the errors for the arrival time estimation task. **left:** Histogram showing the distribution of arrival time errors made by the algorithm against the test set, with a bin width of 0.025 s. **right:** Scatter-plot showing the distribution of errors with respect to SNR.

**Table 1 sensors-19-00597-t001:** Layer Parameters for a single stack of our TCN architecture. Descriptions of the columns are as follows: l represents the layer number within the stack, k represents the kernel size (also known as the filter length or number of weights in each learned digital filter), d represents the dilation rate, Input/Output represent the total number of samples in the input and output sequences, and Receptive Field represents the number of samples in the input sequence ’seen’ by the filters at that layer.

l	k	d	Input	Output	Receptive Field
1	16	2	14,400	14,400	31
2	16	4	14,400	14,400	91
3	16	16	14,400	14,400	331
4	16	256	14,400	14,400	4171

**Table 2 sensors-19-00597-t002:** Decay Rate Optimization.

λ	Recall (α=1%)	MAE
0.015	0.622	0.640
0.020	0.721	0.560
0.040	0.713	0.476

**Table 3 sensors-19-00597-t003:** Algorithm Efficiency by Station. The efficiency of each algorithm (DeepPick, FBPicker and KTPicker) is shown for each of the seven stations for a full year. The first column contains the total number of events found in the corresponding array beam catalog. The subsequent columns contain the detections (true positives), and recall (true positive rate) and false positive rate for each of the algorithms. The last row of the table gives the overall results of each algorithm against the combined catalog across all seven arrays.

	Catalog	DP Picks			FB Picks			KT Picks		
**STA**	**Events**	**TP**	**TPR**	**FPR**	**TP**	**TPR**	**FPR**	**TP**	**TPR**	**FPR**
TXAR	16451	9265	57%	0.001	2933	18%	0.002	6040	37%	0.002
PDAR	12980	6966	54%	0.001	2118	17%	0.003	3691	29%	0.001
ILAR	20769	10269	50%	0.002	3677	18%	0.005	6371	31%	0.002
BURAR	4645	3685	80%	0.001	1565	34%	0.004	2679	58%	0.001
ABKAR	8072	5940	74%	0.002	4015	50%	0.004	5951	74%	0.002
MKAR	40583	24473	61%	0.001	14118	35%	0.002	20031	50%	0.001
ASAR	37595	18204	49%	0.002	9146	25%	0.005	13752	37%	0.003
**TOTAL**	**141095**	**78802**	**56%**	**0.001**	**37572**	**27%**	**0.003**	**58515**	**42%**	**0.002**

**Table 4 sensors-19-00597-t004:** Algorithm Precision by Station. Showing the mean average error (in seconds) for the arrival time estimates of each algorithm. The final row shows the average error across all seven arrays.

STA	DP	FB	b
TXAR	0.447	0.531	0.747
PDAR	0.468	0.487	0.768
ILAR	0.45	0.488	0.69
BURAR	0.477	0.481	0.643
ABKAR	0.384	0.42	0.592
MKAR	0.407	0.443	0.657
ASAR	0.484	0.538	0.692
**TOTAL**	**0.445**	**0.484**	**0.684**

**Table 5 sensors-19-00597-t005:** Algorithm Computational Efficiency by Station. Here we detail the runtime, in seconds, required for each algorithm to process the full year of data at each array.

STA	DP	FB	KT
TXAR	763	22,800	257,800
PDAR	781	18,961	259,243
ILAR	735	19,372	251,210
BURAR	767	22,983	262,368
ABKAR	791	22,913	254,185
MKAR	754	22,838	271,536
ASAR	725	19,059	255,829
**AVG**	**759**	**21,275**	**258,881**

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
