# Peer review of "Improving Regional and Teleseismic Detection for Single-Trace Waveforms Using a Deep Temporal Convolutional Neural Network Trained with an Array-Beam Catalog"

_sensors, 2019, doi:10.3390/s19030597_

Round 1

Reviewer 1 Report

Dear authors,

I took a very careful look in your work. You have done a decent effort in the field of automatic detection of seismic events at regional and teleseismic distances. However, I am not convinced by the way you present your work and this is the reason that I propose major revision.

Comments:

1.      Bibliographic research is limited. Many well-referenced works in the field of automatic detection of seismic events are not taken into account in your work. Please check the bibliography!

2.      The presentation of the DeepPick capabilities in the Appendix A is very poor while you have a very good database of seismic events. Why this?

3.      The Discussion (Part 5) is poorly structured. You have to provide comparison of your results with other algorithms. I mean to implement 2-3 other already published algorithms and to test them in the same database used for DeepPick. Then, please present a comparative graph (DeepPick's detection efficiency versus other detection algorithms efficiency). This is very important in order to convince the readers for the efficiency of DeepPick.

4.      Please, include in the Discussion (Part 5) a paragraph concerning the computational cost of your algorithm.

5.      You have to provide more details in the figure captions concerning the interpretation of your results.

6.      Please, re-organize your abstract and the section of your conclusions, after you implement the pre-mentioned comments.

I wish you good luck

Kind regards

Author Response

Thank you for your thorough review of our manuscript. We believe we have addressed all the revisions pointed out, and the paper is much better because of it. The attached pdf details our point-by-point response.

Reviewer 2 Report

This paper proposed a novel seismic detection algorithm capable of array-like performance from a single trace and utilized temporal convolution neural network to accomplish this task. The problem it addressed is very important in the area of seismic detection and the writing is pretty good. Additionally, I have some comments about this work.

1. The title of the paper is not very suitable. Because the architecture TCN is not proposed by this work. The title should reflect the key novelty of this work.

2. The authors should illustrate training details of the TCN.

3. Line 253, the authors mentioned a parameter number of filters. But the parameter was never mentioned in the architecture before. It is very confusing for readers. Again, readers need more details about the TCN.

4. In section 3.1.1, the authors should explain how to establish high-fidelity sources. Maybe a sketch or a formula could make it clearer.

Author Response

(The authors gave the same response as above.)

Round 2

Reviewer 1 Report

Dear authors,

I read very careful the revised MS (along with the answers to the reviewers) and it is really improved. Based on the following comments I propose minor revision for your work:

1.      Line 62: Reference 9 (Panagiotakis et al. 2008) does not introduce the KTPicker. In addition, please check the format of this reference in the list at the end. It does not correspond to the right format of the Bibliography. You can put this Ref. somewhere in the 1. Introduction or 2. Related work sections.

2.       Lines 512-603. Please check again all references because in some of them the format is not the appropriate. For example see FREIBERGER, W.F. (1963).

Kind regards

Author Response

Dear Editors and Reviewers of the Special Issue on Deep Learning Remote Sensing Data,

Once again, thank-you so much for your kind consideration! The peer-review comments have been incredibly helpful, and the prompt responses have been greatly appreciated. We believe we have implemented all the reviewer comments, as detailed below. Additionally, we have attached the v2 manuscript with all changes highlighted in yellow. Thanks again.

Sincerely,
Joshua Dickey
[email protected]

RESPONSE TO REVIEWER 1 COMMENTS:

Point 1: Line 62: Reference 9 (Panagiotakis et al. 2008) does not introduce the KTPicker. In addition, please check the format of this reference in the list at the end. It does not correspond to the right format of the Bibliography. You can put this Ref. somewhere in the 1. Introduction or 2. Related work sections.
Response 1: We replaced Reference 9 (Panagiotakis et al. 2008) with the appropriate reference (Baillard et al. 2013) in two locations in the body of the manuscript (Line 62 and 383). We also corrected the bibliographic entry (Line 531).

Point 2: Lines 512-603. Please check again all references because in some of them the format is not the appropriate. For example see FREIBERGER, W.F. (1963).
Response 2: We apologize for the formatting of the bibliography… multiple corrections have been made, most notable on Lines 536, 538, 544, 555, 566, and 591.

Reviewer 2 Report

The authors have addressed most of my concerns, and only some minor issues should be corrected, such as typo, description style, etc.